# A Qualitative Study Examining Parental Involvement in Youth Sports over a One-Year Intervention Program

**DOI:** 10.3390/ijerph16193563

**Published:** 2019-09-24

**Authors:** Ausra Lisinskiene, Marc Lochbaum

**Affiliations:** 1Education Academy, Vytautas Magnus University, Kaunas 44248, Lithuania; 2Department of Kinesiology and Sport Management, Texas Tech University, Lubbock, TX 79409-3011, USA; marc.lochbaum@ttu.edu

**Keywords:** parent, child, youth sports, attachment, phenomenological research, intervention study, longitudinal qualitative design

## Abstract

The purpose of this 12-month intervention program was to examine parent–child relationship changes within the sports context. A qualitative interpretative phenomenological analysis was used for the study design. Ten families consented to in-depth interviews. The participants were 10 youth sport parents who had one child each aged 5–6 years. The intervention program involved the participation of all the parents and children. The program integrated psychological, educational, and sports skills into pre-organized sports training sessions. The study results revealed that the intervention program had a positive impact on the parent–child relationship in the sports context. Additionally, the study results suggest that parental involvement in the intervention program positively affected parent–child attachment, the quality of interpersonal relationships between the parent and the child, and effective parenting strategies. Future intervention programs should include both parent and children dyads.

## 1. Introduction

In childhood, parental support is the main and most important influence on the child’s decision to start playing sports [1,2,3,4,5]. Researchers highlight that parent–child relationships depend on the early attachment style formed in the family [6,7]. The attachment of children to parents develops in early childhood, and it is a very important period for creating the foundation of positive close relationships between parents and children in the future [1,8]. Attachment is essential to children’s psychological welfare and forms the basis of personality development and socialization [8]. With our research, we sought to gain a better understanding of how to build parent–child attachment and a close relationship through sport. Thus, to achieve our aims, we grounded our work in Bowlby’s (1969/1982) attachment theory [8,9]. J. Bowlby [8,9] believed that infants are biologically predisposed to form selective bonds with special and proximate caring figures in their environment, proposing that experiences in relation to such bonds are a critical factors in the development of internal working models of the world, the self, and self-in-relation-to-world.

One of the central tenets of attachment theory is the notion that early childhood lays the foundation for the development of personality through the lifespan [7]. Recognizing the psychological importance of children’s relationships with initial caregivers (typically parents) [9,10] hypothesized that children construct internal working models because of initial attachment relationships that will serve for future patterns of cognition, affect, and behavior [7,11]. Researchers have also noted that secure attachment to parents gives children a more secure emotional basis that they can always rely on [7,11,12]. However, based on research, children can be securely or insecurely attached to parents and it directly influence the child and his or her life. In addition to this, it is widely recognized that parents have a greatest influence in many contexts. This research analyzes specifically the sports context. It has been widely recognized that parents can positively or negatively influence youth sports participation. For instance, parents can demonstrate ethical or destructive behavior towards their child, they can act as a positive or negative role models and they can model the overall parent–child relations. Researchers highlight that it largely depends on how parents perceive an understanding of being a youth sports parent. For example, Knight and Holt [13] suggest that optimal parental involvement is achieved when parents strive to both understand and enhance their child’s experience, recognizing that each child is an individual with specific requirements, and that youth sports experiences often occur over an extensive period.

For sports psychology professionals, who work within the youth sports environment, one of the most significant challenges is working with parents’ concerns, formulating positive and realistic expectations for their children’s sporting success [14]. Researchers highlight the need for family-based interventions in the youth sports environment [5]. In this sense, practitioners must understand the complexity of family life and appreciate that every family is unique [15]. The interventions and strategies that work with one family will not necessarily work with other families, and practitioners must be flexible in their approach [16]. Ha, Ng, Lonsdale, Lubans, and Ng [17] highlight that physical activity intervention often only targets children, but interventions that also involve parents may be more beneficial. Ha, Ng, Lonsdale, Lubans, and Ng [17] have developed the “Active 1 + FUN” family-based intervention program which targets the behaviors of parents and children and parents’ parenting methods, which may yield promising results regarding the family’s engagement in physical activity during leisure time. In addition, Baker, and Sanders [18] investigated improvements in child behavior and parenting, following participation in a brief online parenting program (Triple P Online Brief). Baker and Sanders [18] found that higher baseline levels of child behavior problems, older parental age, and more intense conflict over parenting pre-intervention predicted greater improvement in child behavior. Moreover, Bronikowski, Bronikowska, Pluta, Maciaszek, Tomaczk, and Glapa [19] developed a 15-week family focused intervention program, “Juniors for Seniors”, and found a positive impact on physical activity (PA) and health behavior changes. The program was designed to target and improve the perceived sports competence of both children and their parents and to reinitiate or increase the PA of a family unit. The focus was on the development of fundamental bodily skills and sports-specific skills adjusted to the age group. Researchers found increases in PA levels and reductions in sedentary time such as improved sport and motor skills, more frequent family social behaviours (walks, meals, and visiting relatives), or simply improved quality of “do-together” leisure time PA. Laukkanen, Pesola, Finni, and Sääkslahti [20] evaluated a year-long cluster-randomized controlled efficacy trial study and found that parents within the lowest initial parental support intervention tertile significantly increased their support, and their children’s mean level of leisure-time PA significantly improved compared with the corresponding controls during the counseling period. The intervention was found to have an unfavorable influence, especially in the PA of children of initially highly supportive parents. The examples of interventions that are provided above brings important insights in relation to physical activity and in strengthening the parent–child relationship and overall family involvement in the PA of children. However, there is a lack of studies specifically analyzing the parent-child attachment in relation to the sports environment. The integration of both the psychological and educational skills into the sports environment is an inovative step as well.

Based on an understanding of the complexities and sensitivity required, we developed our 12-month intervention program based on Lisinskiene’s [1] work. Lisinskiene [1] emphasized that parental involvement in children’s sport is more important in the early period of sporting life and becomes less appreciable or unwelcome when children gain sporting experience. Also, in the background of parent–child interrelations in sporting activities, adolescents’ alienation from parents is more common than communication and trust.

Again, based on Lisinskiene’s [1] extensive work, her interviews with athlete adolescents revealed that some parents demonstrate substandard behaviour towards their children, such as having no loyalty towards other participants of the sporting activity. Lisinskiene [1] emphasized that the degree and the form of parental involvement in youth sports influence the effectiveness of parent–child relationships. The degree and the form of involvement chosen by the parents are not always appropriate and encouraging, and not always acceptable to adolescents. Parents might benefit from working with sports psychologists to learn coping strategies or psychological skills that could help them manage their emotions in a sports environment and during competitions [1]. In addition to Lisinskiene’s [1] work, the findings of Côté [21], Côté et al., [22], and Fraser-Thomas et al. [23] strongly encouraged us to begin developing this intervention study. Researchers thoroughly interviewed the athletes and found positive developmental experiences related to challenges, meaningful adult and peer relationships, a sense of community, other life experiences, and some of negative developmental experiences (i.e., related to poor coach relationships, negative peer influences, parent pressure, and the challenging psychological environment of competitive sport).

Based on this, Lisinskiene [1] proposed future research that incorporates children from 5–6 years of age to participate in joint classes together with their parents with the purpose of strengthening the parent–child relationship. The overall goal of this study was to develop a one-year intervention program for parents in youth sport to strengthen parent–child interactions. There is a need for applied research to create a better understanding of (1) how to strengthen parental involvement in youth sport and (2) how the long-term intervention could influence parent–child relationship changes. Therefore, we based our work on the integration of psychological (family and individual councelling), educational (theory classes for parents and children), and sports skills (martial arts training sessions) into a year-long program. To the best of our knowledge, our intervention is unique in that it is the first reported intervention where the parent and the child participate together in joint educational, psychological, and sports classes.

## 2. Materials and Methods

### 2.1. Qualitative Research Design

To evaluate the intervention program, we adopted a longitudinal qualitative design. The data analysis complied with the methodological requirements of Interpretative Phenomenological Analysis [24]. The Interpretative Phenomenological Analysis (IPA) method, developed by Smith, Ref. [24] was selected from all available qualitative research methods. IPA has a double purpose: it is both a data collection and qualitative data analysis method. The choice to conduct IPA was reasonable for conducting rich and reflective individual parents’ experiences. Other common focuses for IPA are the trends reflecting both phenomenological and interpretative aspects. We analyzed and interpreted the lived experiences of the research participants. In this sense, we engaged in a double hermeneutic because the researcher is trying to make a view of the participant’s story, and trying to understand and interpretate the individual’s stories. This captures the dual role of the researcher. Interpretative phenomenological analysis focuses on the lived experience of participants by incorporating phenomenology and interpretation. It shares the aims of idiographic phenomenology, which provides a detailed analysis of elements ofthe reflective personal and subjective view of individual experiences. IPA moves one step beyond phenomenology (the participant’s accounts) and attempts to report on the participant’s experience by considering the researcher’s view of the world during interpretation.

### 2.2. Intervention Program

Based on Lisinskiene’s [1] mixed methods study results, a 12-month intervention program on parental involvement in a youth sports program was developed. The program, including psychological, educational, and sports skills training, occurred within a Lithuanian private sports club for martial arts attended by parents and children (see Figure 1). Video taping and the analysis of all sessions occurred with appropriate professionals in sports psychology, sports education, and the taught sports skills, which assisted in evaluating the sessions. Parents participated in practical sports classes together with their child once a week. Outside of the scheduled sports session, both the parent and child scheduled a meeting each month to discuss the program. The program concluded with a skill examination organized by the appropriate Lithuanian Sports Federation for martial arts.

The goal of the psychological aspect of this intervention program was to surround parents with a psychological safe environment and support them in any situation that occurs in sport. More specifically, the goal of the psychological aspect was to analyze the specific parent–child situations that occur in a youth sport environment and discuss these with sport psychologists. The program involved individual and family counseling and personal supervision. For instance, individual counselling included training class analysis which enabled us to analyze each of the family relationships in the class, the behavioral changes in a specific situation, and the reaction of the child, the parent, or the coach in such situations. The training class was discussed individually with parents in a separate individual counseling class. Moreover, the program involved an educational theory class together with educational workshops led by a sports psychologist and sports educator for parents (once in a month/one academic hour) and children (once in a month/one academic hour) to help them understand the benefit of involving children in sport.

The goal of the educational theory class was to include educational skills in influencing positive parent participation in youth sport. The educational classes included seminars and workshops, as well as group discussions. The discussions with other parents during the seminars and interactive participation workshops enabled parents to understand the specific parent behavior and emotion management aspects and helped to them to understand how to behave in a specific situation. For instance, the educational class included a theory class led by the sports educator on the topic of “the overinvolved parent”. During the class, all the group participants watched a video with specific overinvolved parent actions. Together with the sports educator, parents analyzed the video and discussed the topic as a group. Later, the parents offered suggestions to each other on how to manage such a situation. They discussed what specific actions should be taken to avoid such parent behavior. Parents discussed how such situations influenced young athlete behavior and emotions. The parent session themes were as follows: (a) The importance of the parent–child attachment in youth sport; (b) Personality development in youth sport; (c) The importance of the parental involvement in youth sport; (d) Involved/overinvolved/not involved parents; (e) The coach’s role in youth sport parenting; (f) Positive parenting strategies in youth sport; (g) Identifying children’s personal needs in youth sport; (h) Motivation in youth sport; (i) Parent–child educational interaction in sport; parent–coach interaction in sport; (j) Coach–child interaction in sport; and (k) Parent–child–coach educational interaction in youth sport. The child theory class involved drawing sessions and discussions about the same topics as the parents. However, the topics were analyzed through drawing classes, video analysis, and discussions. For example, the parent–child interaction in sport topic was framed around a drawing class with the task to draw positive parent–child relationships in sport. Later, children discussed the topic in a group and explained their drawings.

### 2.3. Participants

The sample consisted of 10 parents (M age = 34), with one child aged 5–6 years. All parents were two parent families. All families had one or two children. In this sense, families varied in relation to siblings. In this research, the child’s birth order was as follows: the youngest child in the family participated in the study. Parents were familiar with the sports club because of the older son’s or daughter’s participation in it. That is exactly how parents were introduced to a new program and decided to participate in it voluntarily. The research participants selection followed the inclusion criteria, information coverage, and informed consent to participate in the survey [17]. The inclusion criteria were as follows: only those parents who had a child aged 5–6 years, and those with a child who is willing to be involved in sports activities were involved. Regarding information coverage, the researchers restricted this to a number of 10 research participants in order to carefully give full attention to each of the parents and each of the children during the research process. More importantly information coverage was based on understanding that IPA requires a detail and in depth examination and attention into every detail of the research process. In this case, 10 parents and 10 children were chosen as a maximum number of research participants in this study. Regarding the sensitive topic of analyzing parent–child relationships in youth sports, the researchers carefully planned the research process in the following stages. First, ethical approval from the first author’s institution, Vytautas Magnus University, was obtained to conduct the research. Next, the researcher organized a meeting with parents and provided a detailed explanation about the ongoing research and the program. Finally, parents consent to participate for their child and for themselves in the study was obtained prior to the planned research. Consent to participate for the child was obtained only from the parent. However interest in participation from the child was respected. Regarding and respecting the child’s needs, it was agreed that if the child demonstrated a lack of motivation to participate in the classes in the program, then the family would not be involved in the program any more and would leave the program. The consent included information about voluntary parent and child participation in the program which is based on research. Consent to participate in the study was received from all the potential participants prior to the research process. The research participants were informed of their right to refuse to participate in the research at any time without any additional explanation. The research participants were informed that they would be guaranteed anonymity and confidentiality. All personal identification information would be removed or changed during the research data analysis and would not be available for third parties. The participants were informed that this research would be conducted in line with all university, legal and local ethical standards.

### 2.4. Data Collection and Procedure

In this study, we obtained data on the parents’ experiences through semi-structured in-depth interviews in three separate time phases: before the intervention, during the intervention, and after the intervention. A semi-structured interview is a suitable method for data collection when the researcher’s priority is to understand a topic in-depth [24]. Therefore, this type of research is primarily focused on insights and understanding. The interview questions were a suggestive guide rather than a mandatory set of questions. Moreover, additional questions, based on the individual stories and situations of a participant, emerged. The interview guide was developed from a Lisinskiene [1] study. The questionnaire consisted of three parts: (1) educational aspects of the intervention study; (2) psychological aspects of the intervention study; (3) sports aspects of the intervention study. These three themes dominated three interviewing processes: before, during, and after intervention.

In this qualitative research, ten stories in three separate time phases (in total 30 interviews) of parents were voice recorded. Interviews were conducted in the first author’s research office, and the schedule was organized in separate individual meetings with parents scheduled in advance at the time convenient for all parties. The interviews lasted from 1 to 1.5 h. The participants in the research were treated as true research experts, and their proposed topics and suggestions were accepted and analyzed in depth.

### 2.5. Data Analysis

The analysis contained the following stages. From the audiotape, we made verbatim transcriptions of each interview [25], which totaled 36 h. At this stage, the focus was on how the participants talked about themselves: their tone, rhythm, pauses, and changes in topics. IPA requires detailing and comprehensive interview transcription material (text), which is the object of the analysis. Therefore, some essential facts of participant interactions are important (laughing, crying, silence, apparent changes in mood, etc.). IPA’s data analysis steps dictated the analysis: reading and rereading; initial noting; developing emergent themes; moving to the next case and looking for patterns across cases. We followed [26] four principles of research quality assessment: (a) sensitivity to context; (b) researcher’s reflexivity; (c) research transparency and consistency; and (d) research relevance and utility. Triangulating the findings by member checking, evaluating inter-coder agreement, rich descriptions of the cases, reviewing and resolving discontinuous evidence, and auditing by peers secured credibility assurance.

## 3. Results

We presented three general categories from the parents’ perspectives for parental involvement in the youth sports intervention program: the parents’ experiences before the intervention (Table 1), during the intervention (Table 2), and after the intervention (Table 3). We collected 3 youth sport parents’ stories. Firstly, parents were asked to express themselves and to tell their experiences before the intervention program. The parents recognized that the decision to start the program was related to feelings such as fear, hope, and desire. Later, the parents talked about the intervention program (after the six months of parents and children attending). The research participants mentioned that the parents’ feelings were more balanced and related to an experience which they had not experienced before, which is related to careful involvement in youth sport together with the child. The parents acknowledged the program’s professionalism and that a significant impact on creating secure parent–child attachment can be provided through sport. The interaction between the psychologist educator and sports professional were reported as an important element in the program in creating secure parental involvement in youth sport. Finally, parental experiences after the program were discussed. The emotional, behavioral, and cognitive changes throughout the 12-month intervention program appeared to be important when evaluating the program. Parent–child and overall family relations and behavior have changed: parents and children spent more time together and family members became actively involved in activities. The emotional parent–child attachment, attachment to the coach and attachment to the sport was highlighted by the parents. Cognitive changes also occurred, such as learning new skills; in the philosophy of sport, a family education course, a new technique was learned. The qualitative results theme table (before the intervention) is presented in Table 1.

### 3.1. Before the Intervention

Psychological aspect: The parents openly discussed the ambiguous feelings that arose before the intervention program. All the parents mentioned that they had never participated in such an organized sports program, which is based on parent and child participation at joint sports classes. This activity resulted in a range of feelings. One feeling was fear, as indicated in the following quotes: “*The decision to participate in a long-term program is a commitment and responsibility, but at the same time, it is important to start trying to help the child to be[come] involved, especially at this young age”* (A1).

A second feeling that emerged was hope. The parents talked about their hope to see their child’s personality development through beginning sport at a young age. The parents also talked about the involvement of their child in sport and the possibility to observe their child and detect their child’s interest or talent. To improve themselves through the program was an important factor for the parents. However, the parents were more determined to participate in the program because of their child, not because of themselves. Thoughts such as these were expressed by the parents before the program. The program tasks and goals were explained clearly. However, they (the parents) did not recognize that the program is more focused on both the parent’s and the child’s personality development and overall family involvement. The main goal for them (the parents) was to do their best from the parent’s perspectives in influencing the child: “*It is mine, as a parent[al] ambition. There is a hope that my child would be safely integrated into youth sport”* (C2).

The parents reported that the integration, detection of the child‘s talent or interest, healthy habits formation and purposeful personality development, as well as enjoyment, were particularly important factors influencing the decision to start the program. The parents wished to integrate their child into a sports environment, finding new friends and building character through sport. To prepare the children for life and to learn strength were the main reasons: “*To start preparing the child for life, to learn new lessons, such as wins and loses through sport and it is the most beneficial environment to do so”* (T4).

*“To be an example for him. Children act the way parents act. I have to be a good example. It is always easier to leave the child to [a] coach, to pay registration and other fees and hope he/she will educate your child. But to put some effort and find time to be involved together with your child is challenging. And I am ready to do it”*.(A1)

Educational aspect: By analyzing and interpreting the study results before the intervention program, for the researcher, it was important to hear the parents’ voices. Regarding the educational aspect, the parents mentioned that the lack of cooperation skills, lack of communication skills, and lack of social skills, particularly in the youth sports context, are the main issues that are related to their own decision to participate as well as their child’s participation: ***“****My son is afraid of the people. I found a serious problem for him to be involved in any activities. I was afraid he wouldn’t enter this program. However, this program was a possibility for him to be safely integrated and to learn to be around people. He entered into a gym because parents were at present” (L6).*

The parents explained that, for their child, it is necessary to learn social skills and to learn communication skills and interact with other children. They believe that such integration would help their child to be socially accepted later at school. The quote presented above by a parent presents and explains the hopeless situation: the parent tried so many activities, which were unsuccessful. The child always left the gym at the very beginning of the training, or other sessions. The parent explained that this program was beneficial because of the joint parent–child classes allowed the formation of a secure attachment between the parent and the child. Parents admitted that the child felt safe and encouraged. Eventually, children became more independent and felt safely integrated. A second feeling that emerged was hope. Parents talked about their hope to see their child’s personality development through beginning sport at a young age. The parents also talked about the involvement of children in sport and the possibility to observe their child and detect their child’s interest or talent. To improve themselves through the program was an important factor for the parents. However, the parents were more determined to participate in the program because of their child not because of themselves. Such thoughts were expressed by the parents before the program. The program tasks and goals were explained clearly. However, they (parents) did not recognize that the program is more focused on both the parent’s and the child’s personality development and overall family involvement. The main goal for them (parents) was to do their best from the parent perspectives in influencing the child: “*It is mine, as a parent ambition. There is a hope that my child would be safely integrated into youth sport”.* (C2). Researchers felt the need for a deeper and careful explanation to parents that the program is designed for the parents and for the children as well. The program focuses on a secure attachment style and healthy relationship formation within the family.

Sports aspect: Most of the parents highlighted that they want to integrate their child into sports due to many reasons: The child previously not engaging in any activities, sports being good for building personality strength, sports being an activity which keeps the child busy, etc. At the same time, all the parents were satisfied to enter a program which involves combined activities that were integrated with sports, educational and psychological aspects. *“There were no sports activity before. I know that sports is good for the personality development. The character develops through the sports. And it is good for a child to be involved in such sports, to learn new skills, to be able to be good at something”* (M9). The parents talked positively about the program as the first activity of this type that they were involved in. They were excited to be part of the program because of the very strong educational aspect of the program: *“I am happy, we will participate in our first activity, and with such a strong educational thing. It not only about the sports as competitive thing, which is good, but also about the overall sports education. Not only the child but also [the] parent is being educated” (A1). “I don’t know what to expect, but I feel that I am in good hands. There were introduction[s] into this program. I can see the professionals work here. So I am quite satisfied” (C2). “It’s because I will find here everything in one place. The sports that my child is crazy about. The consultations, the theoretical classes, workshops. Psychologists that I think is very important. All what is needed. You feel surrounded by a safe environment” (L6).*

### 3.2. During the Intervention

General categories from the parents’ perspectives for parental involvement in the youth sports during the intervention program are presented in Table 2. 

Psychological aspect: During the program, the parents expressed different emotions. The new experience brought parents and children closer together. Family members started to observe changes in family relations. More sport-related topics arose in the family background. The parents were happy participating together in such activities with their children. The parents expressed strong feelings after seeing their child change. The parents reflected that their behavior changes: “*I now see that all the topics are about how we achieved, how we experienced, how we learned. And we learned together, I and my child” (D10)*.

All the parents mentioned new experiences when participating in the program. At times, these experiences were unexpected. Different feelings were expressed by the parents. Some parents mentioned that the program is unique with a very meaningful purpose. Others stated that the program has a strong psychological influence. There were critical moments that the program did not reach its goals because affecting the parents can be a challenging task. As one of the participants noted: “*I came with different thinking. As for me, I was at times declining the information. But they (organizers) somehow attracted my attention. Gave examples during the workshops, and video analysis with a real parent positive and negative behaviors. That was influential. I changed my thinking. I always thought I know better how to educate my child myself. I was wrong” (M9).*

All the parents care about their children, but in very different ways. In sport, in the specific context where the parents, coach, and children are focused on one direction, all three members should understand the same rules in the sports environment to reach successful results. The program was developed to form an honest, communication-based relationship between the parent, the child, and the coach. The program was designed to formate a secure attachment between the members. The parents observed the program. They were the real experts of the program evaluation, as one of the ten participants noted: “*I was observing all the time. I observed how professionals work, how they communicate. I thought: People here work. They work. Very well organized and professional. That is what is needed. All the activities are well planned and have a very deep insight. They mixed up my mind” (M9). “Need to put a lot of effort. During the program, I thought it is final. There is no time. I was about to quit. But the coach planned every activity with a very thoughtful purpose in mind. All activities are planned in a sequence that you automatically are curious and want to see what is going to be next week? So you automatically are in” (G7).*

Educational aspect: During the intervention program, the parents learned new skills. The parents found different communications skills which foster parent–child interaction, parent–parent interaction, as well as parent–coach interaction.

The parents mentioned that during the educational workshops, they learned that there must be a two way communication in order to strengthen the dual role in youth sports. For example, parent–child interaction will be maximized through the open and honest communication where both sides are heard, especially the child, as one of the fathers mentioned: *“It is almost usual that the parent is the boss, and the child should listen to what parent has to say. Being honest, I am just now learning what the child wants to say to me. I am still learning to hear the childs voice, as one of the educators highlighted - it’s the most important in starting building a healthy relationship. And I am still practicing” (M9).*

The new knowledge that is important for parents to be positively involved in youth sports was the most surprising for the parents: *“In youth sports you as a parent need to behave in a different way. There are specific rules here in sports* (smiles)*” (T4). The phrase “parent as an observer” made the biggest impact. And I try to observe. I used to critically comment on specific topics, as I want my child [to be] the best, that’s what I said to lecturers. But I understood that this specific behavior can destroy the relationship and can be understood as destructive and inappropriate method used by the parent” (T4).*

Sport aspect. Conversations with the parents during the intervention program was an opportunity to researchers to analyze the feedback of parents. In relation to the sports aspect, the parents mentioned that this aspect was the most tangible, as you can immediately see the new skills learned. As for the educational and psychological aspect, there is a need for time to see the behavioral changes. Parents saw the children change after obtaining sports-specific skills: children became more confident, brave, and adventurous. New skills have been learned, new techniques related to the martial arts sports field. In particular, the philosophy of martial arts fascinated the most: *“The martial arts has a specific etiquette when the child bows to the gym, bows to the coach, to the opponent, and it is very beautiful. The child learns life skills. Through this they learn respect others. That is very important to me” (D10).* The other parents remembered how they would come back from the gym and exercise at home; they practiced the techniques at home and, more importantly, the parents admitted that they interact through the sports and that better parent–child interaction is experienced: *“We are partners now (smiles)” (C2). We do all those exercises at home, because the child wants to excel. She wants to be prepared for the exam and to gain the white belt, so she is trying hard. And I am the partner, because I know all the technique[s], we learned it together. That is what I am proud of the most” (C2).* The parents talked about the feelings that make them happy to see their child becoming technically proficient. They show their abilities at home, to their friends, and at school. They show their first trophies and medals and it makes them progress. The parents can see a tangible benefit and they explained such feelings during the conversations. They feel that their child as well as they themselves became stronger physically, as well as becoming more stable psychologically, as they became more knowledgeable. Parents also admitted that the child as well as the parents are becoming more and more attached to the sports. They mentioned that the sports are becoming a daily routine, which is framed as a traditional activity in their family.

### 3.3. After the Intervention

General categories from the parents’ perspectives for parental involvement in the youth sports after the intervention program are presented in Table 3. 

Psychological aspect. The stories of all the participants showed emotional pride in their child. The deliberate participation of parents in children’s sports eventually creates an exceptional experience, especially individual emotions, and deep feelings: “*I am proud of my child, how she (daughter) have achieved, it is so important to me that the child is in a very safe environment where professionals are near you” (T4)*. “*I grew as a parent, and as a personality, and I influenced my child through a positive parent example” (D10)*.

Most of the parents highlighted that the attachment between the child and the parent became more visible and the relationship became closer: *“Going together with the child into training classes was amazing. I could feel the “pulse” of my child every time. Of course, I had to plan hard. I had to find the time and force myself sometimes but being together in a sports class created a very close emotional relationship between us” (D10).*

Educational aspect. The parents emphasized that their skills and knowledge were enriched. Their personality grew together with the child. They had the opportunity to interact with other parents. Their circle of friends expanded. However, the most important aspect that the parents highlighted is that through the 12-month intervention program, they gained significant new skills and understanding of what the youth sport parent role is: *“Through theoretical lectures and individual consultations, I gained good experience and educational-psychological support, I never knew such things” (T4). “The program is oriented to[wards] powerful parent education, which in turn influence[s] the child” (D10).*

The participants also talked about the educational video analysis time, which helped parents working in the group to analyze themselves and to model a specific behavior that would be appropriate in youth sports parenting: *“Video analysis during the theoretical lecturers and workshops was helpful. We were given many of the positive and negative youth sports parenting examples, and they are still in my mind. I found myself in those video situations”.* In relation to this quote, it should be mentioned that parent–parent interactions were seen, as they became as a team in many situations. It was nice to see how parents model their behavior, decisions and actions working together as a team.

It was surprising that the parents positively reflected on the program and the educational aspect of the program and mentioned that it was a useful tool in order to determine specific answers to many questions related to overall parenting: *“I can see that I learned not only youth sports parenting but also parenting itself” (C2).*

*“I think that every single sports program should start with this kind of parental involvement concept in order properly to start and realize where we are”*.(A1)

Sports aspect: The parents reflected on themselves after the one-year intervention program. They believe the program strongly influenced their parent–child interactions, emotional attachment, mutual understanding, and overall family interpersonal relations. A positive example is illustrating the changes in parenting life—when they are starting to do sports together with their child. The parents engage in sports through children: *“I can be near my child and spend more time with her. At the same time, it is good for a parent, it is to reorientate myself into a different teammate role” (D10). “The time spent not only at home, but also in training sessions, and in the same gym, enabled to be involved not only myself, my child, but whole family’s attitude towards sports changed” (M9).*

The parents reflects about the specific sports skills learned, but at the same time mentions that the skills and knowledge learned occurred due to being involved in youth sports in a positive way: *“I never knew about the martial arts until now, so now I know karate technique, tactics, history,* etc. *But the most important [thing is] I now know how to feel my child (in sport), where you must be involved and where you must be only an observer. How to be not overinvolved and not dominate because with such behavior parents could damage their own child” (D10).*

The parents were surprised that the program actively involved parents, that it interconnected the parent–child relationship and it interconnected the psychoeducational and sports skills into one single phenomenon. At the end of the program, they (the parents) recognized that the program is oriented towards the child, but through the education of parents. They can reflect and recognize some specific but very important points related to the program: “*We all know that we came here because of our kids. It’s all about them. But now, I can see that it’s about us, parents as well” (A1).*

To change one’s thinking, behavior, or attitude is a difficult task. For us researchers, it was interesting to hear that parents emphasized not only the sport-related habit changes but the overall family habit changes. It was a surprising finding that we could not have even expected: “*I am happy to see my child growing through the sports [providing a] healthy and meaningful activity. This activity became even more meaningful because I was involved in it too. Because I saw, I tried, I felt, I have participated, I evaluated. I saw everything from the inside. It actually changed family habbits, thinking, perceptions” (A1).*

Another example that illustrates the long-term effective change is that the child accepts parent participation and their participation becomes attached. This quote is in line with of the attachment theory, which emphasizes the secure attachment bond between the parent and the child: *“[The] first month of attending was surprising. Being together with the child and wearing the same kimonos was so special. Emotions are high. At the end of the program my thinking changed, I thought, I want to continue. Because Tom (son) says, mom let’s prepare for the training classes, let’s go. We have the same goals the same interest”* (T4). *“My son says: Let’s go, let’s go, we will be late to the trainings. Get your kimono and let’s go. And I was laughing loud. I was so happy to hear such thoughts. And I was like: Okay, it means that it became as a habit, it is our time together, and it is like a natural process to go together to the trainings. That is amazing” (T4).*

## 4. Discussion

The aim of this 12-month intervention program was to examine parent–child relationship changes in sport. A qualitative interpretative phenomenological analysis was used as the study design. A total of 10 youth sport parents took part in semi-structured in-depth interviews. We developed parental involvement into a youth sports intervention program by integrating psychological, educational, and sports skills into the training sessions and qualitative research enabled us to evaluate the program. In total, ten individual interviews with parents were analyzed in three different topics/stages: before the intervention, during the intervention, and after the intervention.

Different topics related to the psychological aspect, such as fear, hope, and desire emerged when parents expressed their experiences before the program. The results revealed that starting the program was challenging for parents. They entered the program with a natural fear, but also a great deal of hope and desire. Lack of confidence, hesitation, and the feeling of uncertainty were the main topics that emerged and emphasized parent fear. The parents explained that they had to find time in their busy schedules. They had to rethink their daily routine. The parents had to decide and make the decision to participate. They had a hope that such a program could help their children to enter the sport and grow in a sports environment. The building of a healthy lifestyle, finding new friends and be social acceptance were also the main topics that emphasized parent desire to enter the program. The parents desired to see their child grow in a safe environment. The development of purposeful personality was also an important factor. As for the educational aspect, before starting the program, the parents acknowledged that they lack knowledge and the specific skills necessary to be a youth sports parent. Theoretical classes and workshops together with other parents enabled them to enrich their overall parent knowledge related to the sports environment. The parents’ hope to integrate their children to be socially accepted, and to learn communication skills and collaboration skills through the sports was one of the most important aspects mentioned. The parents were confident about choosing the field of sports because it was their own child’s decision. Almost all the children wished to participate in martial arts. Other parents kindly invited and offered to try this field of sport with their children. The parents believe that sports develop character and individuality and that is the main reason why they entered this program. The other reason was their curiosity to be involved together with their child and to experience such an activity together. A nice example involving a parent was the wish to help his child to integrate, because the child is afraid of people. The parent was ready to be involved and to help the child to develop social skills. In this sense, this program enabled the child to be safely and carefully integrated.

The parents expressed different experiences when talking about their experiences during the program. All the parents found the program very useful and interacted. The major theme “psychological aspect” emphasizes the parents’ positive emotions felt during the program: enjoyment, parent–child relationship through sports, and parent–child attachment. During the program, the parents could feel the formation of a closer bond between the parent and the child because of the joint sports classes. The parents also mentioned that not only the training classes, but also the educational classes and workshops helped them to understand the specific rules that are related to youth sports parenting and to integrate the learned theoretical skills into practice. Parent–child interaction, safe integration, and enjoyment from seeing their child starting sports were revealed as the experiences obtained by the parents. They mentioned that the activities, such as the theory class for parents and children, and individual consultations helped them to build an understanding of the specific rules that are important in the sports environment. A closer relationship and attachment with their child was formed. Parents acknowledged that the close relationship between the parent and the child was felt and warm feelings appeared. Based on participant stories, the child felt safety and confidence and became self motivated. Parent-child trust, communication, support, and understanding appeared. The interactions between the parent and the child became closer. They began to function as a dual role not only in the sports context but in the daily routine atmosphere. In this sense, Bowlby’s (1969) attachment theory is a useful framework in trying to understand the attachment formation process. The psychological and educational impact was characterized as necessary in a sports environment and enabled the parents to manage their child’s and their own emotions. The parents also mentioned that the educational impact is important in terms of gaining new knowledge and understanding the parent role and behavior in youth sport.

Finally, when the parents talked about how they felt after the 12-month intervention program, the behavioral, emotional, and cognitive changes were identified as the most important changes during the parents’ and childrens’ participation in the 12-month intervention program.

All the parents indicated a positive behavioral change in themselves and their children. The qualitative themes such as personality development changes and moral education emerged and confirmed the major theme of behavioral change. The emotional effect, seen in outcomes such as enjoyment participating in joint sports classes together with their child, seemed to be most important for the parents. The parents also pointed out their pride in seeing their child playing sport and seeing how they developed a purposeful personality through sport. The empathic educational interaction between the parent, the child, and the coach emerged and seemed to be a very important predictor that influenced the decision of the future sports participation of the child and the parent as well. Moreover, the cognitive change was reflected as learned new skills, for example the educational and psychological aspects of sport and of parenting in youth sport. The parents emphasized that knowledge levels increased—both for the parents and child. The theoretical sessions, and individual and family counseling atmosphere created an educational interaction environment in the youth sports setting. In this vein, reference [27] developed a collaborative sports psychology program for parents and presented a series of workshops designed for parents and conducted as part of a larger sports psychology program in a youth sports setting. The researchers focused on psychological parent practical training through interactive, participatory workshops and this enabled a positive impact on overall parenting in youth sports training. Our program was intervention based and focused more on integrating psychological, educational, and sports skills in a youth sports parenting setting. Moreover, Hellstedt’s [28] work presented a case study of an athletic family that illustrates the steps a practitioner might incorporate within a family education approach. In addition, reference [29] presented working with parents in sports model (WWPS Model) and suggested that sports psychologists and coaches create a positive sporting experience for young athletes, which may ultimately enrich and foster more harmonious athlete–parent relationships.

When developing such programs, it is important to enhance parents’ knowledge and understanding of the sports policies. These procedures can enhance communication in the development of effective relationships with coaches, trainers, and management personnel. Increasing parents’ knowledge and understanding can also help to promote and foster positive relationships between parents. Parents are integral to youth sport, and developing relationships between parents can provide a positive yet informal social support network that could contribute to the development of a positive sporting experience for all. Our study results confirmed the importance of the interaction between the sports psychologist, sports educator, and sports coach in strengthening parental involvement in youth sport. Richards et al. [30] developed a one-year formal educational program for parents of competitive gymnasts. The program was focused on educating and enhancing the parents’ ability to create and maintain a task-involved climate for their athletes within a competitive gymnastics environment. Researchers have found that the educational classes were a useful intervention in enhancing parental involvement and parental influence on the athletes’ sport participation and motivation. In this sense, an educational impact on parent involvement is essential in creating a motivational climate in youth sport. Our results confirmed that given the challenges and demands associated with parenting in youth sport, families felt they are understood and supported by the practitioners. Through practical training, theoretical classes, individual counseling, and family counseling, the parents gained a deeper understanding of the parent and family role in youth sport. The parents emphasized that the program made a major contribution not only to the parent–child relationship in sport but enhanced the parent–child relationship at home and their dependency on each other to participate in or to choose sports in the future. As the parents and children spent more time together, the attitude towards each other appeared to change. More related topics to be discussed in the family background arose. The parents mentioned that the spouse relationship changed positively. The mutual understanding, enjoyment, appreciation, and pride were the most significant changes between spouses and strongly affected the child. While evaluating the overall intervention program, it should be mentioned that the participant conversations demonstrated a positive change between the topics before and after the 12 month intervention program. For example, before the intervention program the voice of fear and lack of skills or knowledge appeared. After the intervention program parents together with their children became more confident, resilient, and motivated.This is a positive change that builds a healthy and stable relationships between the parent and the child. The parent self motivation becomes a positive example for the child as well. It also motivates the family to be involved in the activities that enrich the daily routine and brings joy between the family members.

Overall, the parents’ reflections on the program were largely positive. We could state that the intervention program had a positive impact on the psychological, educational and sports aspects, and overall family changes, and strengthened parenting strategies in youth sport and daily family life as well. The parents critically reflected and highlighted the need for educational and psychological skills to be positively involved in youth sport. The need for sports psychologists and sports educators to work along with sports coaches to maximize the benefits of youth sport and youth sports parenting was discussed. Reference [15] highlights that the family experience changes and such change is required for development to occur. The extent to which families negotiate such changes and adapt to the different requirements influences the quality of the interactions within the family.

The uniqueness of this parental involvement in the youth sports program is that the training classes were held together with the child and this created a safe integration for the child into youth sport. The positive educational and psychological interaction between the parent, the child, and the coach emerged and ensured secure integration into the sport and the child’s continued participation in sporting activities. This intervention program can be useful in helping sport psychology, sport education practitioners, and coaches who work with children and parents. This type of program as a new concept may also be an innovative method that can be integrated into a youth sport system and the sports environment. As our study illustarates, having a sport psychologist was a strength of this intervention. If possible, any youth sports systems should include sports psychologist as the main professional who would work along with sports coaches. The interaction between the psychologist, coaches, athletes, and their parents would be valuable. If there is no possibility to do this, coaches could be more involved in educating themselves and raise a bigger challenges for themselves to continuously study and participate in psychological-educational studies, workshops, or seminars. Universities could include topics related to the psychological and educational aspects of analyzing the interpersonal relationships within the family members in the university curricula in order to educate coaches in greater depth. Moreover, community organizations could learn from this current intervention study that it is necessary to involve parents and children together in a community based activities and organize such activities in order to build parent-child attachment and healthy relationships within the smallest, but most important cell—the family.

## 5. Limitations and Directions for Future Research 

This study has three main limitations. First, the attachment styles that were developed in the first years of the studied children were unknown. It would be beneficial to conduct in-depth psychological research in trying to determine particularly secure or insecure attachment styles that were formed in the early years of the children’s lives. Second, the observations as a qualitative strategy would have been valuable insight in this study, however it was not provided. Third, web-based psychological/educational sessions would have been a major strength in adding the overall value to this program in educating parents and coaches. In this sense, the continuity of the program would be ensured with parents and coaches being supported by educational topics, discussions, and tasks. Therefore, even with these few limitations, taken together this intervention study results could be a useful framework for policy makers and practitioners in integrating such a concept in sports and other community-based environments in order to ensure not only a continuously positive relationship between the parent and the child, but also between the members of the community system. Such an environment would ensure a healthy relationship between all of these members.

## 6. Conclusions 

The study results revealed that the intervention program had a positive impact on the parent–child relationship and attachment in sport. The psychological, educational, and sports aspects were observed and had a perceptible impact on parent–child relationship building and overall family relationship changes. The study results revealed that the parental involvement intervention program affected parent–child attachment and the quality of the interpersonal relationships between the parent and the child, and also increased positive parenting strategies.

## Figures and Tables

**Figure 1 ijerph-16-03563-f001:**
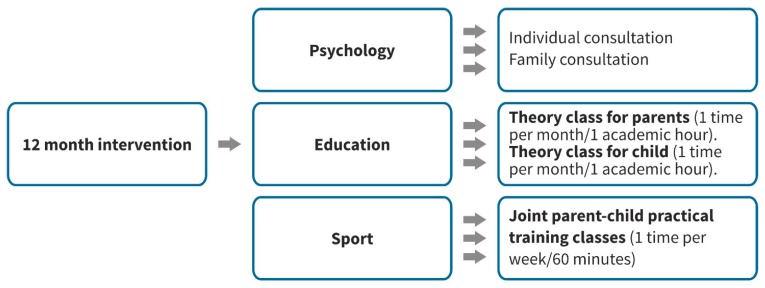
Parental involvement in a youth sport intervention program.

**Table 1 ijerph-16-03563-t001:** Before the intervention.

Super-Ordinary Theme	Themes	Subthemes
Psychology	Fear	Lack of confidence to start the program
		Feeling of uncertainty
		Hesitation
	Hope	Personality development through sport
		To involve child into a sport
		To detect child’s interest
		To improve parents themselves
	Desire	Integration
		Healthy child
		Purposeful personality
		Enjoyment
Education	Lack of skills	Lack of cooperation skills
		Lack of ommunication skills
		Lack of social skills
	Lack of knowledge	Strategies in sports parenting
		Methods used in youth sports parenting
		In how to be positevely involved
Sport	No sports activity before	The need of activity
		Curiosity to play sports
		Child desire to enter the sports
	Desire to integrate the child into sports	Parent decision to integrate the child
		Talent identification
		Parent desire child to grow through sports

**Table 2 ijerph-16-03563-t002:** During the intervention.

Super-Ordinary Theme	Themes	Subthemes
Psychology	Enjoyment	Fun time together
		Positive emotions
		The expression of the personalities
	Parent-child bond through sports	Unexperienced parent-child relations
		Enjoying parent-child relationships in the gym
		Time spent together
	Parent-child attachment	The perception of the need of mutual participation
		Closer parent-child relationship
Education	New skills	The parent-child interaction skills
		The parent-parent interaction skills
		The parent-coach interaction skills
	New knowledge	The perception of careful involvement into childs sport
		A careful communication between the parent and the child
Sport	New activity	Secure child integration into the sport
		Positive parent involvement
	New technique	The ability to be good at something
		The ability to express themselves through the learned technique
	Group processes	

**Table 3 ijerph-16-03563-t003:** After the intervention.

Super-Ordinary Theme	Themes	Subthemes
Psychology	Emotional change	Empowering parenting
		Parent-child motivation
		Family member motivation
		Parent-child-coach motivation
	Cognitive change	Detailed personality analysis
		Changed communication
		Discovered new connection with child
Education	New skills gained	Cooperation skills transfomed into practice
		Communication skills transformed into practice
		Social skills transformed into practice
	New knowledge gained	Positive parenting in youth sports strategies developed
		Perception of positive and negative youth sport parenting arose
Sport	Attachment to sports	Parents start sports
		The child is involved in sports
		Attachment to coach
		Family involvement into sports
		Purposeful leisure time activities
	Dependency on mutual participation	The need of parent-child interaction in sports
		The mutual routne on sports
		The same parent-child interests

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
