# Peer review of "A Qualitative Study Examining Parental Involvement in Youth Sports over a One-Year Intervention Program"

_ijerph, 2019, doi:10.3390/ijerph16193563_

Round 1
Reviewer 1 Report
Overall the paper has been strengthened considerably, that said there are still areas for further work and development. Whilst the grammar and style is markedly improved in the attached I identify several areas where further development and refinement is needed.
My major concern, relates to the results – within the results reference is made to the video recordings and to views of the researcher. Firstly, as it is stated that the video observations are not part of the present article reference to this should be removed as should reference to ‘researcher thoughts’ that cannot be substantiated via results by the reader. Within the results there is also a need to review tense throughout.
Please see attached file.

Author Response
Dear Reviewer,
Thank you very much for your careful revision. A revised version of the manuscript could be found in the attached file.
Thank you.

Reviewer 2 Report
Thank you for the opportunity to review your paper. I thought it was much improved and I appreciate the work and thoughtfulness provided in the response. I do have one more issue that needs to be addresses prior to publication:
Right now your Qualitative Research Design section merges the design and analysis description. I would pull those sections apart. It is difficult to explain the analysis without understanding the data you are collecting, so I would end your research design after page 3, line 25. Then move lines 25-36 to a section at the bottom of methods called Data Analysis.
Author Response
Dear Reviewer,
Thank you very much for the careful revision. A revised version of the manuscript could be found in the attached file.
Thank you.
